# An EPR Study on Highly Stable Nitroxyl-Nitroxyl Biradicals for Dynamic Nuclear Polarization Applications at High Magnetic Fields

**DOI:** 10.3390/molecules28041926

**Published:** 2023-02-17

**Authors:** Nargiz B. Asanbaeva, Sergey A. Dobrynin, Denis A. Morozov, Nadia Haro-Mares, Torsten Gutmann, Gerd Buntkowsky, Elena G. Bagryanskaya

**Affiliations:** 1N.N. Vorozhtsov Institute of Organic Chemistry, 9 Ac. Lavrentiev Avenue, Novosibirsk 630090, Russia; 2TU Darmstadt, Eduard-Zintl-Institute for Inorganic and Physical Chemistry, Alarich-Weiss-Straße 8, 64287 Darmstadt, Germany

**Keywords:** dynamic nuclear polarization, solid state NMR, EPR, biradicals

## Abstract

Nitroxide biradicals are efficient polarizing agents in dynamic nuclear polarization (DNP) solid-state nuclear magnetic resonance. Many recently reported radicals possess substantial DNP efficiency in organic solvents but have poor solubility in water media which is unfavorable for biological applications. In this paper, we report DNP efficiency at a high magnetic field for two water-soluble biradicals resistant to reducing media. Water solubility was achieved by obtaining the radicals in the form of quaternary ammonium salts. Parameters of *hyperfine interaction* and exchange interaction were quantified by EPR spectroscopy, and their influence on the DNP effect was determined. The resistance of the biradicals to strongly reducing media was characterized. High stability was achieved using tetraethyl substituents and pyrrolidine moieties.

## 1. Introduction

Solid-state NMR spectroscopy with magic-angle spinning (MAS) is a powerful method for obtaining essential structural information about biological samples. Nonetheless, the main problem with this method is its low sensitivity, due to the small nuclear spin population difference between ground and excited states in an applied magnetic field. Dynamic nuclear polarization (DNP) [1,2,3,4] is based on the polarization transfer, from highly polarized unpaired electron spins of polarizing agents (PAs) to nuclear spins (e.g., ^1^H or ^13^C) of samples by microwave irradiation. This technique subsequently enhances the NMR signal-to-noise ratio; theoretically, up to 660-fold enhancement may be achieved for ^1^H at a given temperature. PAs that are commonly used to gain the sensitivity of solid-state NMR experiments are biradicals. As mononitroxides or trityls, biradicals are employed to measure physiological parameters—pH [5] and oxygen, nitrogen oxide [6], and thiol concentrations [7]—in biological systems, but in recent decades they have found their main application in DNP NMR.

During the last decade, various groups of binitroxyls were synthesized and used as PAs: TOTAPOL [8,9,10], bcTol [10,11], AMUPols [12,13], SPIROPols [14,15], bTbk [16], TinyPols [17], AsymPols [18]; and as mixed biradicals: TEMTriPols [19], NATriPols [20], BDPA-based compounds [21,22,23,24], TN [25,26,27], and SNAPols [28]. Nonetheless, when working with such compounds, researchers encounter problems with their solubility in water and their resistivity to reductants, which is a crucial factor during biological sample preparation. Reduction of even one part of a biradical by a cell culture medium immediately leads to an inactive PA form and the failure of a cross-effect DNP experiment. For example, TOTAPOL, when utilized for studying native membranes, has shown a loss of DNP enhancement, likely because of reduction in the biradical by free cysteines [29]. Some investigations of the resistance of most popular PAs to reduction have been carried out earlier [13,30]. It has been shown that increasing the intracellular stability of TOTAPOL or AMUPol is possible due to the addition of N-ethylmaleimide removing endogenous biothiols or regeneration of TOTAPOL by oxidation of its hydroxylamine forms. Although the nitroxides added to cells have been N-ethylmaleimide-protected, the stabilization of biradicals by this technique has not improved DNP performance. The stability of nitroxides correlated with the ring size and substituents neighboring the paramagnetic center [31]. In this paper, we study biradicals which have a skeleton based on five-membered-ring nitroxides and tetraethyl substituents that provide additional stability.

According to some reports, the highest DNP enhancement can be achieved due to the relatively strong exchange interaction (*J*) between the paramagnetic fragments in such biradicals [32]. Nonetheless, recent studies showed that the key factor for significant cross-effect DNP is not the great sum of exchange interaction and dipolar coupling (*D*), but the relative ratio *J*/*D* [33]. In addition, as shown earlier [34,35,36], DNP performance strongly depends on substituents around the radical center and on electron spin relaxation times. For biradicals with longer electron spin-relaxation times, higher DNP enhancement is observed. Thus, in many cases nitroxide biradicals with spirohexyl substituents possess high DNP enhancement. However, it was shown that when spirohexyl substituents do not provide accessibility of the solvent to the radical center, it leads to a decrease in DNP efficiency [36].

The aim of the present study is to continue the research into the development of effective polarization sources by determining structure–DNP performance relationships. Towards this goal, we investigated the application potential of biradicals **3** and **4** (Figure 1) in heteronuclear direct polarization experiments. In particular, the behavior of these biradicals in reducing media was analyzed, parameters of exchange interaction were measured, and the radicals were studied by DNP-enhanced solid-state NMR experiments to examine their efficiency at producing signal enhancement.

## 2. Results and Discussion

The synthetic procedures for radicals **2** and **4** have been previously described [37].

### 2.1. EPR Measurements and Simulations

To characterize the local structure of paramagnetic parts of the novel biradicals, we carried out room temperature (r.t.) X-band (9.87 GHz) continuous-wave EPR experiments on 100 μM solutions in methanol or phosphate-buffered saline (PBS). All biradicals showed a multiplex splitting pattern in the solution, resulting in the superposition of hyperfine coupling to ^14^N of the nitroxides (A_N iso_) and the exchange interaction (*J*) between the two unpaired electrons. The observed X-band EPR spectra of all compounds are shown in Table 1 (column 2). Calculated EPR spectra were fitted to the exchange Hamiltonian expressed as *J·S*_1_*·S*_2_. The obtained parameters (exchange coupling *J* and isotropic hyperfine A_iso_) determined from the fitting analysis are listed in Table 1 (column 3).

*Nitroxide-nitroxide biradicals*. The EPR spectrum of the symmetric rigid biradical **3** can be fitted to A_N iso_ ≈ 43.7 MHz, |*J*| = 206 ± 2 MHz (mean ± SD), whereas for the asymmetric biradical **4**, simulations taking into account one value of exchange interaction failed. It should be noted that for similar biradicals, the authors of [36] have proposed two interpretations of the continuous-wave EPR spectrum. The first explanation is that the biradical is characterized only by zero mean exchange-coupling *J*, but a large width of the exchange-coupling distribution. This interpretation appears to be most probable, and the authors of [36] supported it by pulse EPR experiments. The second way to describe this data set requires an assumption of a considerable proportion of monoradicals present in the sample. We conducted a quantitative analysis of paramagnetic particles in the biradicals **3** and **4** relative to standard calibrated TEMPO and Finland trityl solutions by the CW EPR method. As a result of this experiment, it was determined that the concentration of paramagnetic spins in the solution corresponds to the assumption that all particles in the solution are in the form of biradicals. Therefore, we followed the first pathway of spectra interpretation, and good agreement between calculated and experimental EPR spectra was achieved by taking into account a *J* distribution with the center at *J* = 0 MHz and *σ* (*J*) = 120 MHz.

### 2.2. W-Band Echo-Detected Spectra

Pulsed EPR experiments at W-band (94 GHz) microwave (MW) frequency were acquired on a Bruker Elexsys E680X EPR spectrometer. In the experiments, π/2 pulse lengths of 16 ns were achieved. The experiments were carried out at a temperature of 80 K. ED spectra are shown on Figure 2.

### 2.3. Relaxation Measurements

The spin–lattice relaxation time of the biradicals was measured by the inversion-recovery method (Appendix A). The phase relaxation time of the samples was measured by means of primary echo decay. The obtained traces (Figure 2) were fitted by mono-exponential functions. The results of the fitting are presented in Table 2.

### 2.4. DNP Experiments

In Figure 3a,c, the ^1^H MAS spectra obtained for radicals **3** and **4** in frozen glycerol-d_8_/D_2_O/H_2_O (60:30:10 *v*/*v*/*v*) matrix are shown. The peak at 6.2 ppm is assigned to the isotropic signal from water present in the matrix. For radical **3,** an enhancement of 13 corresponding to a time saving factor of ca. 170 is obtained, which is slightly higher than the value of 9 observed for radical **4**. In the ^1^H→^13^C CPMAS spectra (Figure 3b,d), signals at 60.7 and 70.5 ppm are observed, which are attributed to the aliphatic carbons of glycerol (also part of the matrix). An enhancement of 15 is achieved for radical **3** and an enhancement of 9 is obtained for radical **4**. The similar enhancement factors obtained in the ^1^H→^13^C CPMAS and their corresponding ^1^H MAS spectra clearly indicate a homogenous distribution of the two radicals in frozen solution.

Build-up curves for radicals **3** and **4** in glycerol-d_8_/D_2_O/H_2_O (60:30:10 *v*/*v*/*v*) matrix are shown in Figure 4. Mono-exponential functions were used to fit the data points for both samples. For radical **3** (Figure 4a) a build-up time (T_B_) of 1.10 ± 0.02 s and for radical **4** (Figure 4b) a T_B_ of 0.78 ± 0.03 s are calculated. The slightly shorter T_B_ obtained for radical **4** is in good agreement with the slightly smaller enhancement in the ^1^H MAS spectrum obtained for radical **4** compared to radical **3**.

To investigate the polarization transfer in ^13^C MAS NMR, spectra of biradicals **3** and **4** in glycerol-d_8_/D_2_O/H_2_O (60:30:10 *v*/*v*/*v*) matrix, measured with and without microwave irradiation with a build-up of 300 s, were recorded. To distinguish between direct and indirect polarization transfer, two sets of spectra were measured as described in the literature [38]. The results are shown in Figure 5. The indirect polarization pathway from electron to ^1^H and then via cross-relaxation to ^13^C in the µW_OFF_ spectra (Figure 5b,d) is almost negligible for radicals **3** and **4**. This observation is expected since the polarization transfer through cross-relaxation in solids is limited [39] and most of the time visible only via DNP enhancement. In the µW_ON_ spectra obtained for radical **3** (Figure 5a), the polarization transfer via the ^1^H spin reservoir (indirect path) is clearly visible while for radical **4** (Figure 5c) there is no significant indirect path obtained. This indicates the presence of local motions in the frozen solution of radical **3,** while for radical **4** they are less pronounced. These differences are probably induced by the molecular structure of the radicals, an observation discussed in detail in our previous works [27,40].

Next to the indirect path for both radicals in frozen solutions the direct polarization transfer from electron to ^13^C is observable. For these experimental data sets, an enhancement of 12 is obtained for radical **3** corresponding to a time saving factor of 144, while a value of 11 is reached for radical **4**.

If we compare the radicals, the amplification levels in the ^1^H and the ^1^H→^13^C CPMAS spectra for radical **3** are larger than those for radical **4**. A probable reason is that biradical **3** almost completely consists of a structure with *J* ≈ 200 MHz, in contrast to biradical **4**, which corresponds to the distribution of conformations centered at *J* ≈ 0 MHz. As demonstrated earlier, the presence of conformers with *J* ≈ 0 leads to a decrease in DNP efficiency [38]. In the ^13^C MAS spectra, we observed comparable enhancements for radicals **3** and **4** for direct polarization transfer.

For both biradicals, the observed DNP enhancements are lower than for spiro substituted biradicals presented in [36]—an observation which is probably related to the electron spin relaxation time. To probe this assumption, we measured electron spin relaxation times at 9.7 T for biradicals **3** and **4**. We obtained similar values of ~1 μs (T_2e_) and ~0.3 ms (T_1e_) for biradicals **3** and **4** at a concentration of 0.1 mM at 80 K (Table 2). These values are at least two times shorter compared to spiro-substituted PAs [34,35,36]. Furthermore, we performed our DNP experiments with much higher biradical concentrations of 15 mM, where even shorter values of T_2e_ and T_1e_ are expected. These results are in good agreement with the authors’ earlier work [41], where it was shown that spin relaxation times for radicals containing tetraethyl substituents are shorter compared to the corresponding radicals containing spiro substituents.

Another probable important factor for the obtained DNP efficiency is the shielding of the radical center from solvents by tetraethyl substituents. Kubicki et al. [36] noted that for tetraethyl-containing biradicals, the optimum saturation factor (T_1e_T_2e_) can be reached, but the DNP enhancement is still low, which they explained by radical shielding. The structure of our tetraethyl nitroxides **3** and **4** obtained by X-ray [42,43] clearly shows that this “shielding” arises due to tetraethyl substituents, which form a dense shell at a distance of ~3 Å around the nitroxide moiety. On the other hand, it is this shielding that ensures tetraethyl nitroxide’s high stability against reduction by ascorbate or any other reducing agent, and allows their use in living systems and in the cells [44]. To underline this hypothesis, for our biradicals **3** and **4**, the resistance to reduction by ascorbic acid was inspected (for details see Section 2.3) showing that **3** and **4** are stable for several hours, in contrast to spiro-substituted PAs, which exist for only ~10 min in the presence of ascorbate [36].

### 2.5. Resistance of the Biradicals to Reduction by Ascorbate

To determine the rate of reduction of the biradicals, we added 0.1 M ascorbate at pH 7.4 to 0.3 mM radicals and measured the total signal intensity by EPR spectroscopy as a function of time. The integrated EPR signal reflects a total radical concentration, which, in the absence of reduction, is double the sample concentration. Because the two nitroxide moieties of **3** and **4** can be reduced independently, for precision, we reported this parameter as the total nitroxide concentration. The kinetics of radicals’ **1**–**4** decay in the presence of 0.1 M ascorbate at pH 7.4 and are shown in Figure 6.

It is known that for highly stable nitroxides, the kinetics of reduction are complicated [45] and determined not only by the rate of nitroxide reduction but also by an equilibrium with hydroxylamine and the presence of a nascent ascorbate anion. Thus, to prevent the accumulation of the ascorbate anion, we chose the same approach as in the paper by Bobko et al. [43], and 2 mM glutathione was added into the solution. The initial part (60–120 min) of the kinetics of the decay was approximated to a mono-exponential function to calculate pseudo-first-order rate constants *k_obs_* (Equation (1)):(1)$$I\left(t\right)=A_{0}\cdot e^{-k\cdot_{obs}\cdot t}$$

After that, these constants *k_obs_* were divided by the value of concentration of ascorbic acid to determine second-order reaction constants ***k***. All reduction rate constants of the radicals are presented in Table 3. The rate constants were extracted from the initial part of the kinetics (60–120 min) when the contribution of indirect reactions was negligible [45].

As follows from the obtained data, the most stable radical is **1**. As one can see in Table 3, the reduction in biradicals **3** and **4** is limited by the reduction in the most stable paramagnetic moiety **2** and **1**, respectively.

In Figure 7, readers can observe a change of EPR spectra of biradical **4** in the presence of ascorbate. With time, the contribution of the monoradical part (three curves, black arrows) increases and the exchange lines (two curves, red arrows) degrade. Moreover, magnetic-resonance parameters of the formed mononitroxide correspond to those of radical **1.** This observation is consistent with the finding that radical **2** is less stable than radical **1** (Table 3).

## 3. Materials and Methods

### 3.1. EPR Sample Preparation for W-Band EPR Experiments

EPR samples for W-band measurements of **3**–**4** were dissolved at a concentration of ~100 μM in a mixture of deuterated DMSO and methanol (at 20:30, *v*/*v*) immediately before the experiments. Approximately 2 μL aliquots of these solutions were placed into quartz W-band EPR tubes (0.6 mm inner diameter, open at one end), which were shock-frozen in liquid nitrogen prior to insertion into the precooled cavity. The solvent mixtures used resulted in transparent, glassy samples upon rapid freezing.

### 3.2. X-Band EPR Experiments

CW electron paramagnetic resonance (EPR) spectra were recorded on a commercial Bruker (9 GHz) spectrometer, Elexsys E 540 (Bruker Corporation, Billerica, MA, USA), in water (~10^−4^ M for **1**–**4**) solutions in PBS, at pH 7.4, at room temperature. The EPR spectra were obtained with the following spectrometer settings: frequency, 9.87 GHz; microwave power, 0.5 mW; modulation amplitude, 0.05–0.15 mT for biradicals **1**–**4**; time constant, 20.5 ms; and conversion time, 20 ms. For kinetic measurements, the EPR spectra were performed at the same parameters but with a higher modulation amplitude (0.05–0.15 mT) to optimize the signal-to-noise ratio. Simulations of solution ESR lines were acquired in the EasySpin software 5.2.28 [46] (accessed on 16 May 2020), which is available at http://www.easyspin.org.

In order to measure reduction kinetics, stock solutions of a nitroxide radicals, ascorbic acid, and glutathione were prepared in phosphate buffer (10 mM, pH 7.4), and pH 7.4 was achieved with NaHCO_3_. All components were carefully and quickly mixed in a small tube to obtain required concentrations (radical: 0.3 mM, GSH: 2 mM, and ascorbate: 10–100 mM), and were inserted into an EPR capillary (50 μL). The capillary was closed on both sides and placed into the EPR resonator. The decay of integral intensity of the EPR spectrum was monitored to characterize the kinetics.

### 3.3. Solid-State DNP Experiments

#### 3.3.1. Sample Preparation

For solid-state DNP experiments, 15 mM solutions of each binitroxyl radical (**3** and **4**) were prepared using a mixture of 60%*v* of glycerol-d_8_, 30%*v* of D_2_O, and 10%*v* of H_2_O as the matrix. Approximately 25 µL of each solution was packed into 3.2 mm sapphire rotors. To prevent any leakage, a silicon plug was placed between the liquid sample and the ZrO_2_ drive cap.

#### 3.3.2. DNP Experiments

All solid-state DNP experiments were performed on a Bruker Avance III 400 MHz NMR spectrometer equipped with an Ascend 400 sweep-able DNP magnet and a 3.2 mm triple resonance ^1^H/X/Y low-temperature MAS probe. The microwave (µW) was provided by a 9.7 T Bruker gyrotron system operating at 263 GHz. Measurements were conducted at nominally 115 K in the presence (µW_ON_) and absence (µW_OFF_) of microwave irradiation. The ^1^H MAS, ^1^H→^13^C CPMAS, and ^13^C MAS spectra were acquired at a spinning rate of 8 kHz.

To obtain ^1^H MAS spectra, a background suppression pulse sequence was used employing π/2 pulses with 2.3 µs and π pulses with 4.6 µs lengths [47]. A total of 16 scans were accumulated, and a relaxation delay of 4 s was applied.

The ^1^H→^13^C CPMAS experiments were performed with 128 scans, a recycle delay equal to 1.3·T_B_, where T_B_ is the build-up time for ^1^H, and a contact time of 2 ms. Spinal 64 heteronuclear proton decoupling was applied during data acquisition [48].

To obtain the build-up curves for ^1^H, the saturation–recovery pulse sequence was applied. This sequence starts with a saturation pulse train of twenty π/2 pulses with 2.3 µs length and 200 µs pulse spacing, followed by a waiting time to build up magnetization and a π/2 detection pulse. For each data point, 16 scans were accumulated. These measurements were performed with microwave irradiation (µW_ON_).

The ^13^C MAS direct polarization experiments were performed according to reference [49]. In all experiments, a saturation pulse train of fifteen π/2 pulses, each pulse with a length of 3.5 µs and a space of 5 ms between them was applied to the ^13^C channel to have a defined initial magnetization. To select the direct polarization pathway, π pulses with a length of 6 μs with a spacing of 50 ms between them were applied to the proton channel to suppress polarization transfer via proton spin reservoir during the 300 s of build-up of the carbon magnetization. After that, a π/2 pulse of 3.5 µs was applied to the ^13^C channel for detection. In contrast to the direct polarization pathway, the indirect polarization pathway cannot be detected separately. Thus, the indirect pathway was separated by subtracting the spectra obtained with π pulses on the ^1^H channel (showing only the direct path) from the spectrum obtained without π pulses on the ^1^H channel (showing the direct + indirect path). All experiments were carried out with (µw_ON_) and without (µw_OFF_) microwave irradiation for both radicals. A recycle delay of 0.5 s was used. During data acquisition, high-power proton decoupling was performed using the spinal 64 sequence [48].

To calculate the DNP enhancement factors (*ε*), the areas of the peaks obtained with μW_ON_ in the ^1^H MAS, the ^1^H→^13^C CPMAS and the ^13^C MAS spectra, respectively, were divided by the corresponding peak areas obtained with μW_OFF_.

## 4. Conclusions

An EPR study on two highly stable nitroxyl-nitroxyl biradicals was conducted, and their resistance to reduction by ascorbate in a buffer at pH 7.4 was characterized. The stability of the obtained DNP agents was very high due to tetraethyl substituents and pyrrolidine moieties. This, in principle, allows them to be used in cell biological experiments for several hours, which was not feasible with spiro-substituted biradicals. The observed DNP enhancement coefficients were substantial but not outstanding. One possible reason for these results are nonoptimal values of spin exchange interaction owing to flexible linkers. The second explanation is the fast electron spin relaxation induced by the tetraethyl environment. Therefore, it is necessary to strike a balance between stability in biological media and a high DNP coefficient.

## Figures and Tables

**Figure 1 molecules-28-01926-f001:**
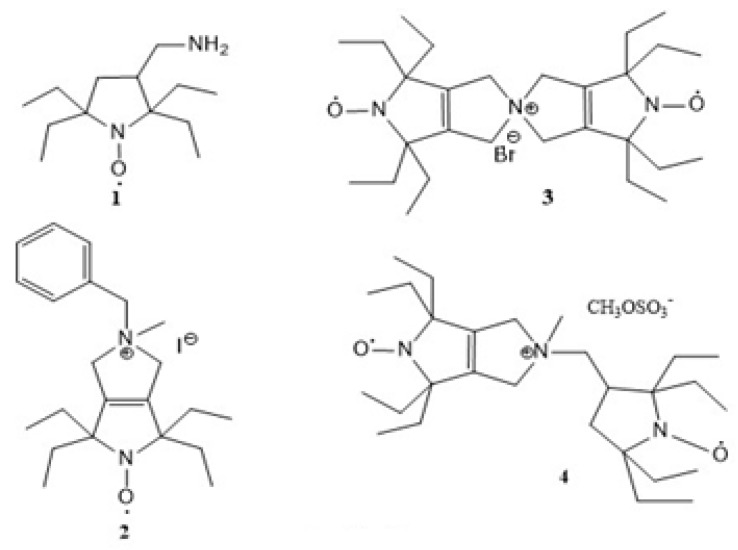
Chemical structures of monoradicals **1** and **2** and binitroxyl radicals **3** and **4**.

**Figure 2 molecules-28-01926-f002:**
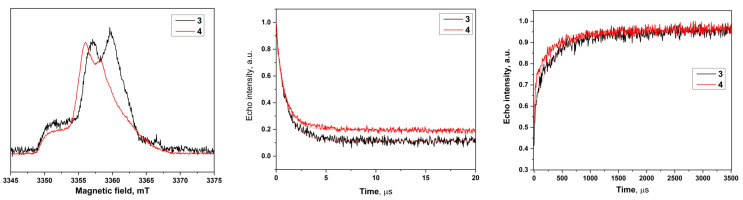
**Left**: The 94 GHz echo-detected EPR spectra (black: for **3**, red: for **4**) at 80 K. **Center**: Intensity of spin echo of biradicals **3**–**4** at 80 K as a function of the time. **Right**: Inversion recovery time traces of biradicals **3**–**4** at 80 K along with mono-exponential fits.

**Figure 3 molecules-28-01926-f003:**
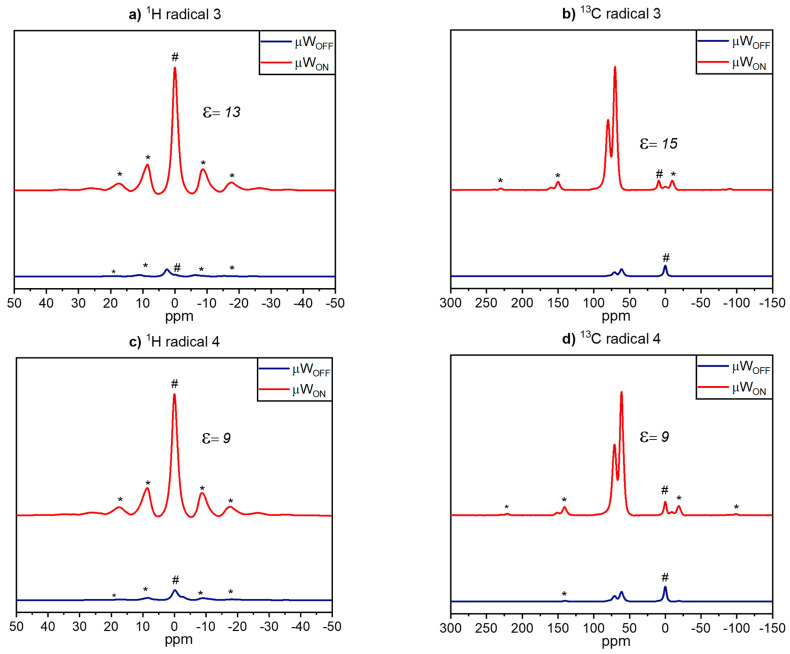
^1^H MAS and ^1^H→^13^C CPMAS spectra of frozen solutions of 15 mM binitroxyl radicals **3** (**a**,**b**) and **4** (**c**,**d**) in glycerol-d_8_/D_2_O/H_2_O (60:30:10 *v*/*v*/*v*). Spectra were recorded with (μW_ON_) or without (μW_OFF_) microwave irradiation at nominally 115 K. The # indicates the signal from the silicon plug near 0 ppm and the asterisks are attributed to spinning sidebands.

**Figure 4 molecules-28-01926-f004:**
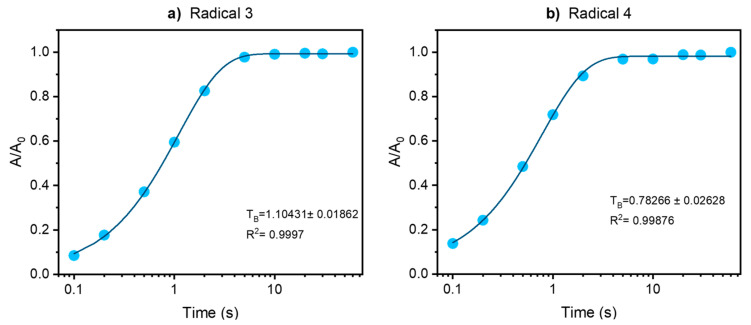
^1^H build-up curves obtained for frozen solutions of radical **3** (**a**) and **4** (**b**) in glycerol-d_8_/D_2_O/H_2_O (60:30:10 *v*/*v*/*v*). A mono-exponential function was used to fit the data points.

**Figure 5 molecules-28-01926-f005:**
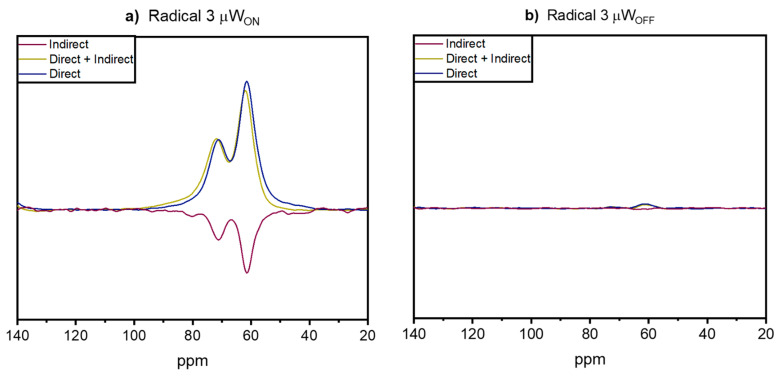
A ^13^C MAS spectra of frozen solutions of 15 mM binitroxyl radicals **3** and **4**, respectively, in glycerol-d_8_/D_2_O/H_2_O (60:30:10 *v*/*v*/*v*) obtained with 300 s as build-up. (**a**) Radical **3** measured with MW irradiation (MW_ON_), (**b**) radical **3** measured without MW irradiation (MW_OFF_), (**c**) radical **4** with MW irradiation (MW_ON_), and (**d**) radical **4** without MW irradiation (MW_OFF_). Spectra were recorded at nominally 115 K.

**Figure 6 molecules-28-01926-f006:**
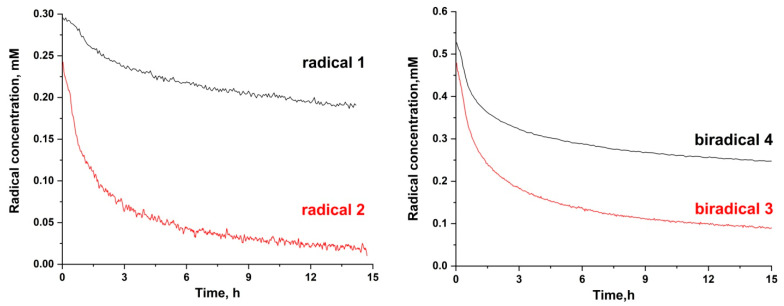
Kinetics of decay of radicals **1**–**4** in the presence of 0.1 M ascorbate in PBS at pH 7.4.

**Figure 7 molecules-28-01926-f007:**
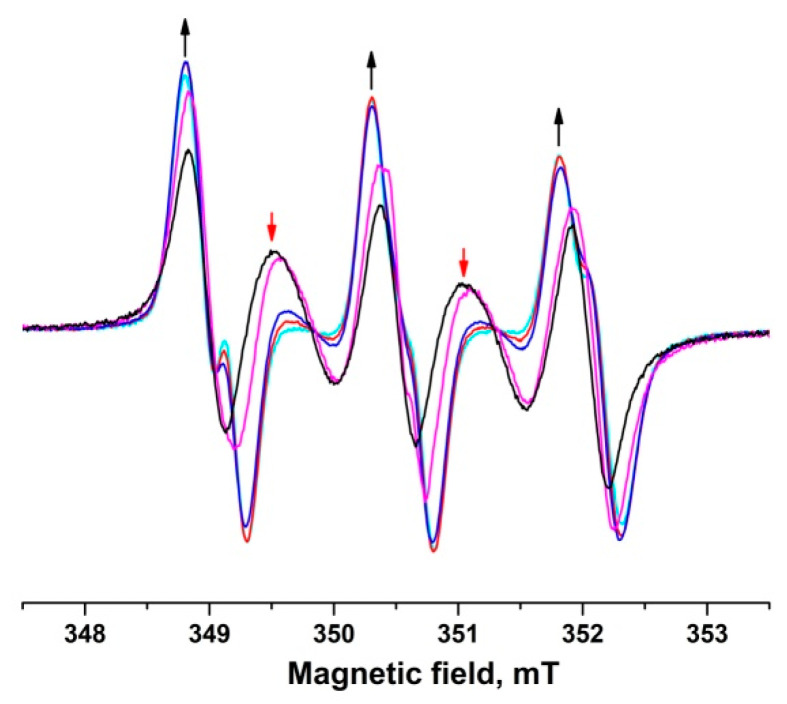
Kinetics of the decay of nitroxide biradical **4** in the presence of 0.1 M ascorbate in PBS at pH 7.4.

**Table 1 molecules-28-01926-t001:** R.t. X-band EPR spectra of ~0.1 mM radicals **1**–**4**.

Radical	EPR Spectrum	Parameters of Simulations
**1**	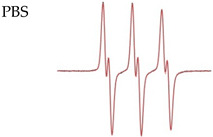	A_N iso_ = 43.0 ± 0.2 MHz A_H iso_ = 7.2 ± 0.2 MHz
**2**	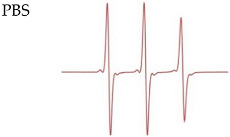	A_N iso_ = 43.7 ± 0.2 MHz
**3**	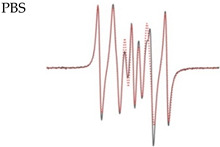	A_N iso_ = 43.7 ± 0.2 MHz *J* = 206 ± 2 MHz
**4**	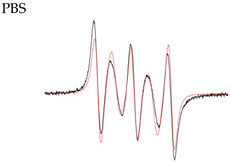	A_N1 iso_ = 44.0 ± 0.2 MHz A_N2 iso_ = 43.2 ± 0.2 MHz *J* = 0 *σ(J)* = 120 MHz *J* distribution

**Table 2 molecules-28-01926-t002:** Parameters of relaxation times for biradicals **3**–**4**.

Biradical	T_2_, μs	T_1_, ms
**3**	1.01 ± 0.01	0.31 ± 0.01
**4**	0.92 ± 0.01	0.29 ± 0.01

**Table 3 molecules-28-01926-t003:** Rate constants of reduction by ascorbate for radicals **1**–**4** (for details, see the experimental part).

Radical	k_obs_ × 10^3^, min^−1^	k × 10^3^, M^−1^s^−1^
**1**	1.4 ± 0.1	0.23 ± 0.02
**2**	11.6 ± 0.3	1.93 ± 0.05
**3**	9.7 ± 0.2	1.62 ± 0.03
**4**	6.2 ± 0.2	1.03 ± 0.03

## Data Availability

Not applicable.

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
