# Peer review of "An EPR Study on Highly Stable Nitroxyl-Nitroxyl Biradicals for Dynamic Nuclear Polarization Applications at High Magnetic Fields"

_molecules, 2023, doi:10.3390/molecules28041926_

Round 1

Reviewer 1 Report

The article analyze the properties of two water soluble and highly stable binitroxyl radicals as potential polarizing agents for DNP NMR. The authors have characterized the biradicals by EPR spectroscopy, analyzed their resistance to reduction by ascorbate and determine their efficiency to polarization transfer by 1H MAS, 13C MAS and 1H-13C CPMAS. Even if the determined DNP enhancement coefficients for the studied biradicals are not high enough to be interesting for this application, the work has been well planned and executed and the article will be interesting for the general reader of this publication.

However, there is a point that should be addressed before publication. For the biradical 4, the EPR spectrum is interpreted in terms of a zero mean exchange coupling with a wide distribution of J values, as previously stated for a similar biradical (JACS-2020). Does this fact imply that for some conformations the interaction is ferromagnetic (J>0) and for others antiferromagnetic (J<0)? In addition, when compound 4 is reduced by ascorbate, it is observed that some lines disappear (the exchange ones) while others gain intensity (those corresponding to a monoradical). Have the EPR spectra of different preparations been recorded to verify that the relative intensities of all lines are exactly invariable, and thus rule out the possible presence of monoradicals?

Author Response

Reviewer 1:

The article analyze the properties of two water soluble and highly stable binitroxyl radicals as potential polarizing agents for DNP NMR. The authors have characterized the biradicals by EPR spectroscopy, analyzed their resistance to reduction by ascorbate and determine their efficiency to polarization transfer by 1H MAS, 13C MAS and 1H-13C CPMAS. Even if the determined DNP enhancement coefficients for the studied biradicals are not high enough to be interesting for this application, the work has been well planned and executed and the article will be interesting for the general reader of this publication.

 However, there is a point that should be addressed before publication. For the biradical 4, the EPR spectrum is interpreted in terms of a zero mean exchange coupling with a wide distribution of J values, as previously stated for a similar biradical (JACS-2020). Does this fact imply that for some conformations the interaction is ferromagnetic (J>0) and for others antiferromagnetic (J<0)? In addition, when compound 4 is reduced by ascorbate, it is observed that some lines disappear (the exchange ones) while others gain intensity (those corresponding to a monoradical). Have the EPR spectra of different preparations been recorded to verify that the relative intensities of all lines are exactly invariable, and thus rule out the possible presence of monoradicals?

Reviewer is right. In case of a zero mean exchange coupling with a wide distribution of J values for some conformations the interaction is ferromagnetic (J>0) and for others antiferromagnetic (J<0). A similar approach to describe the distribution of the exchange interaction was applied in our previous work [ Asanbaeva, N.B.; Gurskaya, L.Y.; Polienko, Y.F.; Rybalova, T. V; Kazantsev, M.S.; Dmitriev, A.A.; Gritsan, N.P.; Haro-Mares, N.; Gutmann, T.; Buntkowsky, G.; et al. Effects of Spiro-Cyclohexane Substitution of Nitroxyl Biradicals on Dynamic Nuclear Polarization. Molecules 2022, 27, doi:10.3390/molecules27103252.]. The relative intensities of all lines in EPR spectra are constant at different concentrations. As you can see, Figure 5 shows a change in the intensity of the lines in the spectrum of the EPR of the biradical 4. Exchange lines (2 and 4) decrease, and lines 1, 3 and 5 ("monoradical") increase. But also in “monoradical” lines the hfi constant on the proton, which was present in the spectrum of monoradical 1 (Table 1), begins to manifest. This suggests that a fragment of the biradical similar to nitroxyl 2 is restored, and the spectrum of a fragment similar to radical 1 begins to mix for the main spectrum of the biradicals.

Reviewer 2 Report

The manuscript by Asanbaeva et al. presented a case study on biradicals and their stability in ascorbic acid solution at pH 7.4. The manuscript contains a full study and descent conclusion. Therefore, I suggest accept after minor revision.

concerns:

The figure captions needs attentions and corrections.

Some pargraphas are loosely written and not scientifically corrected ones.

one example is paragraph, line-85-97

Why the radical 2 has highest Kobs? authors must comment on it.

What is the reason for the CP/MAS signals even at µOFF? authors must comment on it.

Figure 5: the traces should be labelled in the figure itself

Author Response

The manuscript by Asanbaeva et al. presented a case study on biradicals and their stability in ascorbic acid solution at pH 7.4. The manuscript contains a full study and descent conclusion. Therefore, I suggest accept after minor revision.

concerns:

The figure captions needs attentions and corrections.

Reviewer is right. We corrected the figure captions and formatted them in the same style

Some paragraphas are loosely written and not scientifically corrected ones.

one example is paragraph, line-85-97

We agree with reviewer and add the following sentences:

“We conducted a quantitative analysis of paramagnetic particles in the biradicals 3 and 4 relative to standard calibrated TEMPO and Finland trityl solutions by CW EPR method. As a result of this experiment, it was determined that the concentration of paramagnetic spins in the solution corresponds to the assumption that all particles in the solution are in the form of biradicals. Thus, we followed the first pathway of spectra interpretation and good agreement between calculated and experimental EPR spectra was achieved by taking into account a J distribution with the center at J = 0 MHz and σ (J) = 120 MHz.”

Why the radical 2 has highest Kobs? authors must comment on it.

The radical 2 has highest Kobs, because it is less stable radical than other radicals due to difference in nitroxide ring structure. It is known that pyrrolidine type radicals are more resistant to reduction than pyrroline type nitroxides due to the absence of a pi bond in the ring [ Morris, S.; Sosnovsky, G.; Hui, B.; Huber, C.O.; Rao, N.U.M.; Swartz, H.M. Chemical and Electrochemical Reduction Rates of Cyclic Nitroxides (Nitroxyls). J. Pharm. Sci. 1991, 80, 149–152, doi:https://doi.org/10.1002/jps.2600800212.]

What is the reason for the CP/MAS signals even at µOFF? authors must comment on it.

The CP/MAS experiments use polarization from the 1H spin reservoir of the sample and transfer it to a less sensitive X nucleus in our case the 13C which is then observed. Since our samples contain a large number of 1H spins (we used a matrix containing 10% of water) the amount is sufficient to generate a signal in 1H MAS NMR and also in 1H/13C CP MAS NMR even when no microwave irradiation is applied. In case we applied microwaves the polarization of the 1H spin reservoir is enhanced and thus we obtain stronger signals in the MW on compared to the MW off spectra. Since this is a generally accepted observation in DNP NMR experiments, we think that no further comments on this is neccessary.

Figure 5: the traces should be labelled in the figure itself

The kinetic traces for each radical were labeled on Figure 5.